# Regulation of Glycosylation in Bone Metabolism

**DOI:** 10.3390/ijms25073568

**Published:** 2024-03-22

**Authors:** Kazunori Hamamura, Mayu Nagao, Koichi Furukawa

**Affiliations:** 1Department of Pharmacology, School of Dentistry, Aichi Gakuin University, 1-100 Kusumoto-cho, Chikusa-ku, Nagoya 464-8650, Japan; 2Department of Biomedical Sciences, Chubu University College of Life and Health Sciences, Kasugai 487-8501, Aichi, Japan

**Keywords:** glycosylation, glycosphingolipids, glycoprotein, glycosyltransferase, osteoblasts, osteoclasts

## Abstract

Glycosylation plays a crucial role in the maintenance of homeostasis in the body and at the onset of diseases such as inflammation, neurodegeneration, infection, diabetes, and cancer. It is also involved in bone metabolism. *N*- and *O*-glycans have been shown to regulate osteoblast and osteoclast differentiation. We recently demonstrated that ganglio-series and globo-series glycosphingolipids were essential for regulating the proliferation and differentiation of osteoblasts and osteoclasts in glycosyltransferase-knockout mice. Herein, we reviewed the importance of the regulation of bone metabolism by glycoconjugates, such as glycolipids and glycoproteins, including our recent results.

## 1. Introduction

Glycosylation is a biochemical process that adds sugars to lipids and proteins by various glycosyltransferases. It not only plays a crucial role in the development and maintenance of organs and tissues [1,2,3,4] but is also involved in the pathogenesis of disease [5,6,7,8,9,10,11,12,13]. Furthermore, glycosylation has been shown to contribute to bacterial and viral infections [6,7,8]. The hemagglutinin spike in the influenza virus is essential for its adhesion to the sialic acid-containing sugar chain receptors of host cells, and the neuraminidase spike with sialidase activity is critical for virus budding from host cells [7]. The knockout of globo-series Gb3 in mice revealed that verotoxin in *Escherichia coli* O-157 specifically binds to Gb3 in the human digestive tract and kidneys and damages epithelial cells [8]. Among muscular dystrophies, Fukuyama congenital muscular dystrophy accompanied by abnormalities in the central nervous system and muscle–eye–brain (MEB) disease are caused by structural abnormalities in the *O*-mannose-type sugar chain on α-dystroglycan [9].

Glycosylation is also involved in neurodegeneration and regeneration [10,11]. Gangliosides have been shown to regulate complement systems and maintain the integrity of nervous tissues [10]. Moreover, the heparan sulfate of glycosaminoglycans (GAGs) accelerated the regeneration of neuronal axons, whereas the chondroitin sulfate of GAG exerted the opposite effect [11]. 

Glycosylation plays an important role not only as a marker for the detection and monitoring of cancer but also in the acquisition and regulation of malignant properties [5,12,13]. We demonstrated that differences in glycosylation positively or negatively regulated malignant properties in various cancers, such as melanoma, osteosarcoma, and oral squamous cell carcinoma [12,14]. The disialyl-gangliosides GD3 and GD2 promoted malignant properties in melanoma cells, whereas the monosialyl-ganglioside GM1 attenuated these properties [12]. GD3 and GD2 expressed in melanomas and osteosarcomas up-regulated the phosphorylation of focal adhesion kinase, p130Cas, and paxillin by activating the Src family kinases Yes or Lyn, thereby promoting malignant properties [12]. Furthermore, we reported that the blood group antigen Lewis y, which has core 2 (Galβ1-4GlcNAcβ1-3Galβ1-4Glc) structures with two fucosylations, was expressed in oral squamous cell carcinoma cells [14]. Moreover, Lewis y on epidermal growth factor receptor (EGFR) disturbed the binding of EGF to EGFR, thereby inhibiting the phosphorylation of EGFRs [14]. Therefore, modifications by glycosylation regulate the malignant properties of cancer cells.

We recently reported that bone metabolism was regulated by glycosphingolipids. The expression of globo-series Gb4 in osteoblasts promoted their proliferation and increased bone formation. Furthermore, the deletion of GM2/GD2 synthase (β1,4-*N*-acetylgalactosaminyltrasferase: *B4galnt1*) in mice reduced bone formation by decreasing the number of osteoblasts. However, the deletion of GD3 synthase (α2,8-sialyltrasferase: *St8sia1*), which plays a crucial role in the generation of b-series gangliosides, mitigated age-related bone loss in mice [15]. 

In this review, we introduce recent findings on the regulation of bone metabolism by glycoconjugates, such as glycolipids and glycoproteins. 

## 2. Modes of Glycosylation

Glycoconjugates comprise glycoproteins, glycolipids, and proteoglycans (Figure 1).

Glycoproteins are covalently bound sugar chains and proteins, and there are two types based on the binding mode of the sugar chain: *N*- and *O*-linked glycans. In *N*-linked glycans, *N*-acetylglucosamine (GlcNAc) is linked to the amide group of asparagine (Asn) residues. All *N*-glycans have the common core sugar chain sequence, Man α1–6 (Man α1-3) Man β1-4GlcNAc β1-4GlcNAc β1-Asn-X-Ser/Thr, and are classified into three types: oligomannose, only mannose binds to the core sugar chain; complex, GlcNAc binds to the core sugar chain; and hybrid, mannose binds to Man α1–6 of the core sugar chain, while GlcNAc binds to Man α1-3 of the core sugar chain. In *O*-linked glycans, *N*-acetylgalactosamine (GalNAc) is mainly linked to the hydroxyl group of serine (Ser) or threonine (Thr) residues (*O*-GalNAc glycan). Besides *O*-GalNAc, there are *O*-Fuc (α-Fuc), *O*-Glc (β-Glc), *O*-Xyl (β-Xyl), *O*-Man (α-Man), and *O*-GlcNAc (β-GlcNAc). Proteoglycans are composed of core proteins and GAG chains. A GAG is a straight heteropolysaccharide comprising a repeating disaccharide that consists of an amino sugar (GlcNAc and GalNAc) and uronic acid (glucuronic acid and iduronic acid). Glycolipids are bound sugar chains and lipids that are classified as glycosphingolipids and glycoglycerolipids. In humans and mice, glycosphingolipids are superior to glycoglycerolipids in terms of quantity and diversity. Glycosphingolipid strains are divided according to the sugar attached to lactosylceramide [1,16] (Figure 2). The addition of sialic acid to lactosylceramide via an α2,3 linkage produces GM3, which serves as a starting point for ganglioside synthesis. Moreover, the addition of galactose to lactosylceramide via an α1,4 linkage produces Gb3, which serves as a starting point for the synthesis of globo-series glycosphingolipids. GlcNAc is also added to lactosylceramide via a β1,3 linkage to form lacto-/neolacto-series glycosphingolipids.

## 3. Regulatory Roles of Glycosphingolipids in Bone Metabolism

### 3.1. Expression of Glycosphingolipids in Osteoblasts and the Glycosphingolipid-Mediated Regulation of Osteoblast Proliferation and/or Differentiation

Osteoblasts express globo-series Gb4 (globoside) and the a-series gangliosides GM3 and GD1a. The expression levels of these glycosphingolipids decrease after the induction to mature osteoblasts. To investigate whether Gb4 and GD1a expressed in osteoblasts are involved in bone formation, bone phenotypes were analyzed using Gb3 synthase (α1,4-galactosyltransferase: *A4galt*)-knockout mice without Gb4 and GM2/GD2 synthase (β1,4-*N*-acetylgalactosaminyltrasferase: *B4galnt1*)-knockout mice without GD1a. A previous study demonstrated that Gb3 synthase-knockout mice had a lower bone mass than wild-type mice [17]. Calcein double labeling was performed to assess bone formation, and the findings obtained showed that it was lower in Gb3 synthase-knockout mice. In addition, decreases were observed not only in the number of osteoblasts on cancellous bone but also in the mRNA levels of osteoblast markers, including Runt-related transcription factor 2 (Runx2), a key transcription factor for osteogenesis and osteocalcin, in Gb3 synthase-knockout mice (Figure 3). The proliferation of osteoblasts that differentiated from bone marrow cells was also lower in Gb3 synthase-knockout mice than in wild-type mice [18]. On the other hand, no significant differences were observed in the number of osteoclasts on cancellous bone or in the mRNA levels of osteoclast markers, such as tartrate-resistant acid phosphatase (TRAP) and the nuclear factor of activated T-cells (NFATc1), a key transcription factor for osteogenesis, between knockout and wild-type mice. These findings indicate that Gb4 increased bone mass by promoting bone formation. 

In our previous study, the deletion of GM2/GD2 synthase in mice reduced bone formation by decreasing the number of osteoblasts [19]. We also showed that osteoblasts derived from GM2/GD2 synthase-knockout mice did not express GD1a; however, they expressed higher levels of GM3 than osteoblasts derived from wild-type mice. The lack of GD1a expression and increase in GM3 expression may have contributed to reductions in the number of osteoblasts. 

When miglustat, D-threo-1-phenyl-2-decanoylamino-3-morpholino1-propanol (D-PDMP), and D-threo-1-phenyl-2-hexadecanoylamino-3-morpholino1-propanol (D-PPMP), inhibitors of glucosylceramide synthase (GCS), which is responsible for the generation of all glycosphingolipids, were applied to the mouse osteoblast cell line, MC3T3-E1 and the expression of Gb4 and GD1a decreased in osteoblasts and their proliferation was suppressed [20]. Angiopoietin-like 6 (Angptl6) has been shown to play a role in regulating osteoblast proliferation. To establish whether the expression of Gb4 in osteoblasts increases their proliferation, we administered Gb4 to MC3T3-E1 cells in which Gb4 expression levels were reduced by D-PDMP. The findings obtained showed that the administration of Gb4 to D-PDMP-treated osteoblasts increased phosphorylated extracellular signal-regulated kinase 1/2 (p-ERK1/2) and promoted the proliferation of osteoblasts [18]. Consistent with these findings, the proliferation of MC3T3-E1 cells was suppressed following the knockdown of Gb4 synthase [18]. On the other hand, the administration of Gb3, the precursor of Gb4, did not affect osteoblast proliferation. 

Although the mechanisms by which glycosphingolipids regulate the proliferation and differentiation of osteoblasts remain unclear, glycosphingolipids were shown to control signaling for proliferation and differentiation by interacting with receptors in glycolipid-enriched microdomain (GEM)/lipid rafts [3,12,21] (Figure 4). Therefore, dynamic changes in the composition of glycosphingolipids in osteoblasts appear to play an important role in their proliferation.

### 3.2. Expression of Glycosphingolipids in Mesenchymal Stem Cells (MSCs) and Regulatory Roles of Glycosphingolipids in the Differentiation of MSCs to Osteoblasts

MSCs express GM3, GM2, GM1, GD1a, and GD3 [22,23,24,25,26]. The expression level of GD1a was found to increase during the differentiation of MSCs to osteoblasts, whereas that of GM3 decreased [22,24]. Moreover, a treatment with GD1a or the knockdown of the GD1a synthesis gene (St3gal2) using its shRNAs enhanced the differentiation of MSCs to osteoblasts [22,24]. However, the differentiation of MSCs to osteoblasts was inhibited by a GM3 treatment [22]. Previous studies examined the regulatory mechanisms of gangliosides for the differentiation of MSCs to osteoblasts [22,24]. EGFR activation promoted the differentiation of MSCs to osteoblasts [22]. Kim et al. showed that the differentiation of MSCs to osteoblasts was promoted more by a treatment with GD1a and EGF than with EGF alone, whereas it was suppressed more by a treatment with GM3 and EGF than with EGF only [22]. These findings suggest the importance of gangliosides in the regulation of EGF signaling during the differentiation of MSCs to osteoblasts.

The *Lmna* gene encoding Lamin A/C has been shown to regulate the differentiation of MSCs to osteoblasts [27]. In comparisons of MSCs derived from *Lmna* mutant mice and wild-type mice, the differentiation of the former to osteoblasts was lower than that of the latter. Moreover, the levels of the osteoblast differentiation markers, bone morphogenetic protein-2 (BMP-2) and osteocalcin, and GD1a expression were lower in *Lmna* mutant mice than in wild-type mice. The treatment of MSCs from *Lmna* mutant mice with GD1a increased BMP-2 and osteocalcin levels and restored reductions in osteoblast differentiation. 

GCS was found to affect the differentiation of MSCs to osteoblasts or adipocytes [28]. The expression levels of GCS decreased during osteoblastogenesis and increased during adipogenesis. Its knockdown in MSCs using its shRNAs promoted osteoblastogenesis and suppressed adipogenesis through the regulation of peroxisome proliferator-activated receptor γ. These findings demonstrated the important role of gangliosides in regulating the differentiation of MSCs to osteoblasts. 

### 3.3. Expression of Glycosphingolipids in Osteoclasts (Pre-Osteoclasts) and Regulatory Roles of Glycosphingolipids for Differentiation to Osteoclasts

Pre-osteoclasts express GM3, GD1a, and the b-series gangliosides GD3 and GD2. Previous studies reported reductions in their expression levels following the induction of osteoclastogenesis. GCS inhibitors and knockdown using its shRNAi reduced the number of receptor activator of nuclear factor-kappa B (RANKL)-induced TRAP-positive multinucleated osteoclasts [29,30,31]. The administration of lactosylceramide restored D-PDMP-induced decreases in the number of TRAP-positive multinucleated osteoclasts by increasing p-ERK1/2 and I kappa B (IκB) levels [29]. GM3 was also shown to promote osteoclastogenesis [31].

GD3 synthase (α2,8-sialyltransferase: *St8sia1*) plays an important role in the production of b-series gangliosides, and its deletion in mice mitigated age-related bone loss [15]. A previous study reported that b-series gangliosides were essential for the secretion of leptin, and their deletion reduced its secretion by adipocytes [32]. Moreover, leptin promoted bone resorption by osteoclasts via the sympathetic nervous system [33]. Therefore, we hypothesized that a deficiency in b-series gangliosides in mice may suppress bone resorption by reducing the secretion of leptin. No significant differences were observed in bone mass between 15-week-old wild-type mice and their knockout mice, whereas bone mass was greater in 40-week-old knockout mice than in wild-type mice [15] (Figure 5A). This suppression of age-related bone loss in knockout mice was attributed to the inhibition of bone resorption through decreases in the number of osteoclasts (Figure 5B).

## 4. Regulatory Roles of Glycoproteins in Bone Metabolism

### 4.1. Glycoprotein-Mediated Regulation of Osteoblast Proliferation and/or Differentiation

Modifications to *O*-linked β-*N*-GlcNAc (*O*-GlcNAc) have been shown to regulate osteoblast differentiation [34,35,36]. *O*-GlcNAc modifications in global proteins, including Runx2, were promoted during osteoblast differentiation [34,36]. The treatment of MC3T3-E1 cells with ST060266, an *O*-GlcNAc transferase (OGT) inhibitor, or siRNAs for OGT under the induction of osteoblastogenesis decreased the mRNA levels of alkaline phosphatase, osteocalcin, and bone sialoprotein [36]. Moreover, increases were noted in osteocalcin expression in MC3T3-E1 cells under the induction of osteoblastogenesis in the presence of PUGNAc, an inhibitor of *O*-GlcNAcase (OGA) [34]. Bioinformatics experiments revealed that *O*-GlcNAcylation interacted with [Ca^2+^]_i_ and promoted osteoblast differentiation [37]. Therefore, *O*-GlcNAc modifications in global proteins, which are key transcription factors for osteogenesis, increased during osteoblast differentiation, which further promoted their differentiation [34,35,36].

Nagel et al. identified *O*-GlcNAc-modified proteins in differentiated osteoblasts with tandem mass spectrometry using *O*-GlcNAc-modified peptides enriched by wheat germ agglutinin lectin weak affinity chromatography [35]. TGF-β-activated kinase 1/MAP3K7-binding protein 2 (TAB2) and c-AMP response element-binding protein (CREB)-binding protein (CBP), involved in osteoblast functions, were identified [35,38,39,40]. TAB2 binds to and activates TGF-β-activated kinase 1 (TAK1), which then up-regulates the phosphorylation of Runx2 through p38 mitogen-activated protein kinase (MAPK) signaling, resulting in the differentiation of osteoblasts [38]. Moreover, CBP formed a complex with CREB, which activated Runx2 through 3-phosphoinositide-dependent kinase 1 [39]. Runx2 was also identified as an *O*-GlcNAc-modified protein by tandem mass spectrometry [41]. 

Insulin receptor substrate-1 (IRS-1) was modified by *O*-GlcNAc at Ser residues. IRS-1 has been reported to play roles in osteoblast differentiation and the bone anabolic function of parathyroid hormone [42,43]. The treatment of MC3T3-E1 cells with PUGNAc increased IRS-1 Ser^1011^ GlcNAc [44]. Increases in UDP-*N*-acetyl-alpha-*D*-galactosamine:polypeptide *N*-acetylgalactosaminyltransferase 7 also promoted the glycosylation of nephronectin, an extracellular matrix protein, thereby enhancing osteoblast differentiation [45]. 

Protein modification with sugar chains by not only Runx2 but also other markers of osteoblast differentiation leads to the activation of protein functions and osteogenesis. *N*-linked glycosylation sites (Asn-Xaa-Ser/Thr) at the N135 position were necessary for the secretion of BMP-2 and osteoblast differentiation [46]. Sialyation at the termini of both *N*- and *O*-glycans on bone sialoprotein enhanced osteoblast differentiation and affected mineralization through the activation of ERK [47]. The glycosylation site S^89^ of dentin matrix protein 1 was shown to be essential for osteogenesis [48]. These findings demonstrate that the glycosylation of key proteins for osteogenesis plays important roles in their functions.

### 4.2. Regulatory Roles of Glycoproteins for the Differentiation of MSCs to Osteoblasts

The MSC glycome was compared to the glycome of osteoblasts differentiated from MSCs using mass spectrometry and nuclear magnetic resonance spectroscopy [49], and the findings obtained showed that high-mannose structures were more dominant in MSCs. On the other hand, complex *N*-glycans lacking *N*-acetylneuraminic acids and neutral hybrid-type *N*-glycans were enriched in osteoblasts differentiated from MSCs [49]. An analysis of *O*-glycan profiles between MSCs and osteoblasts differentiated from MSCs revealed the enrichment of fucosylated acidic structures in MSCs [49]. A previous study reported that *N*- and *O*-glycan processing in the Golgi regulated the early steps of osteoblast differentiation from MSCs [50]. A treatment with the mannosidase I inhibitor, kifunensine, which completely suppresses *N*-glycan processing in the Golgi, promoted the differentiation of MSCs to osteoblasts by inhibiting the phosphoinositide-3-kinase (PI3K)/Akt pathway [50]. On the other hand, treatment with the mucin-type *O*-glycan synthesis inhibitor benzyl-O-GalNAc suppressed their differentiation [50]. Therefore, modifications to proteins, such as *N*- and *O*-glycans, may play important roles in the regulation of osteogenesis from MSCs. 

### 4.3. Regulation of the Differentiation in the Osteoclasts by Glycoproteins

*O*-GlcNAcylation has been reported to regulate osteoclast differentiation [51,52,53,54]. RANKL increased global *O*-GlcNAc-modified protein levels in macrophages derived from bone marrow, with peaks being observed 2 days after the treatment [53]. A treatment with the OGT inhibitor OSMI-1 and the knockdown of OGT using its shRNAi decreased global *O*-GlcNAc-modified protein levels and reduced the number of TRAP-positive multinuclear cells [54]. The inhibition and knockdown of OGT also reduced NFATc1 levels. Li et al. demonstrated that *O*-GlcNAcylation was required during the early stages of osteoclast differentiation, while a reduction in *O*-GlcNAcylation was needed for the formation of mature osteoclasts [54]. The pharmacological inhibition of OGT suppressed osteoclast differentiation by decreasing NFTc1 levels. On the other hand, the pharmacological inhibition of OGA did not affect NFATc1 levels, but suppressed osteoclast functions, such as bone resorption [54]. Furthermore, nucleoporin 153 (Nup153) was identified as a protein with increased *O*-GlcNAcylation during the early stages of osteoclastogenesis, and *O*-GlcNAcylation was only detected at T546 and S548 in Nup153 [54]. Nup153 has been identified as a component of nuclear pore complexes and regulates the nuclear transport of MYC proto-oncogene, a bHLH transcription factor (MYC) [55,56]. Consistent with these findings, the knockdown of Nup153 using siRNA in RAW264.7 cells reduced the nuclear translocation of MYC in the presence of RANKL and TNFα [54]. The knockdown of Nup153 also decreased the expression of NFATc1 and suppressed the number of TRAP-positive cells [54]. Collectively, these findings demonstrate that *O*-GlcNAcylation in Nup153 plays an important role in osteoclast differentiation. 

*N*-glycosylation has also been reported to regulate osteoclast differentiation [57,58]. Dendritic cell immunoreceptor (DCIR), a member of the C-type lectin receptor family, suppresses osteoclastogenesis [57]. It binds to an asialo-biantennary *N*-glycan (NA2) on bone marrow macrophages [57]. The elimination of the terminal sialic acids of *N*-glycans by a neuraminidase increased the exposure of NA2, which suppressed osteoclastogenesis [57]. Osteopontin on osteoclasts was also found to be *N*-glycosylated at the 79-position asparagine [58]. The overexpression of osteopontin in RAW264.7 cells increased the mRNA levels of NFATc1, cathepsin K, and TRAP, and promoted the nuclear translocation of nuclear factor-kappa B (NF-κB) [58]. However, the overexpression of the N79 glycosylation site of mutant osteopontin did not affect osteoclast-related genes or the nuclear translocation of NF-κB [58]. 

Sialylation plays an important role in osteoclastogenesis [59,60,61,62,63,64,65,66,67]. Mouse bone marrow macrophages and RAW264.7 cells express alpha (2,3)-linked sialic acid and alpha (2,6)-linked sialic acid [59]. A treatment with sialidase and the knockdown of ST6Gal-I revealed the involvement of alpha (2,6)-linked sialic acid in osteoclast differentiation [59]. Sialic acid-binding immunoglobulin-like lectin 15 (Siglec-15) has been reported to regulate the development of osteoclasts [60,61,62,63,64,65,67,68]. Monocytes/macrophages derived from mice in which Siglec-15 was deleted showed an impaired ability for osteoclast differentiation [62]. Siglec-15 expression was induced 2 days after RANKL treatment in bone marrow-derived macrophages [60]. Siglec-15 interacted with DNA polymerase III subunit gamma/tau (DNAX)-activating protein 12 (DAP12), an adaptor protein bearing an immunoreceptor tyrosine-based activation motif, through Lys-272 in Siglec-15 and Asp-52 in DAP12 [60,61]. Siglec-15 increased binding to Syk in a manner that was dependent on the interaction between Siglec-15 and DAP12 in the presence of vitronectin [60]. Therefore, the interaction between Siglec-15 and DAP12 plays an important role in the regulation of functional osteoclast formation. Siglec-15 has been implicated in the RANKL-induced activation of the PI3K and MAPK pathways [61]. The binding of sialylated Toll-like receptor 2 (TLR2) to siglec-15 was shown to initiate cell fusion for osteoclast formation [66]. The sialylation of TLR2 was caused by the sialyltransferase ST3Gal1, which was induced by RANKL [66]. Furthermore, the sialylation status of immunoglobulin G was reported to affect osteoclast differentiation [69]. The treatment of desialylated immune complexes in human preosteoclasts increased the number of osteoclasts and promoted bone resorption [69]. The interaction between desialylated immune complexes and the FcR common γ chains, FcγRII and III, played a key role in enhancing osteoclast differentiation [69].

## 5. Future Scope of Glycoscience in Bone Research

We propose two directions for glycoscience in bone research, the first of which is an investigation of the differentiation tropism of MSCs due to differences in glycosphingolipids expressed in MSCs (Figure 6). MSCs differentiate into osteoblasts, chondrocytes, or adipocytes. If it is possible to identify the glycosphingolipids that promote osteoblast differentiation among those expressed in MSCs, MSCs that are more likely to differentiate into osteoblasts may be isolated by using glycosphingolipids as markers, thereby generating more osteoblasts, which will be useful for the treatment of bone defects. The second direction is to use glycosphingolipids as a tool to elucidate the mechanisms underlying the proliferation and differentiation of osteoblasts. Runx2 in immature osteoblasts and osteocalcin and ATF4 in mature osteoblasts are markers for the differentiation stage of osteoblasts. However, since they are expressed intracellularly, immature and mature osteoblasts cannot be isolated as living cells. If it is possible to identify glycosphingolipids as specific cell surface markers at the differentiation stage of osteoblasts, they will be useful as a tool for clarifying the mechanisms by which osteoblasts proliferate and differentiate.

## 6. Conclusions

Glycosylation is essential for the proliferation and differentiation of osteoblasts and osteoclasts. However, in contrast to other fields, such as cancer research, the role of glycosylation in bone metabolism remains unclear. Further studies are warranted to elucidate the regulatory mechanisms for bone metabolism through the fusion of glycobiology and bone research. 

## Figures and Tables

**Figure 1 ijms-25-03568-f001:**
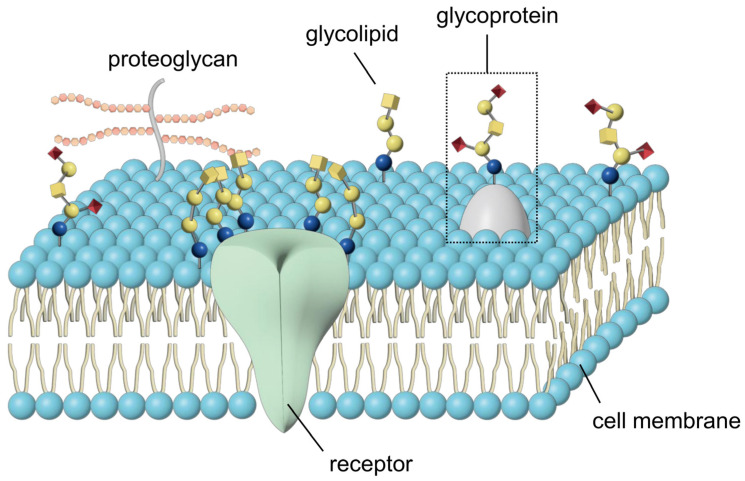
Glycoproteins, glycolipids, and proteoglycans in plasma membrane.

**Figure 2 ijms-25-03568-f002:**
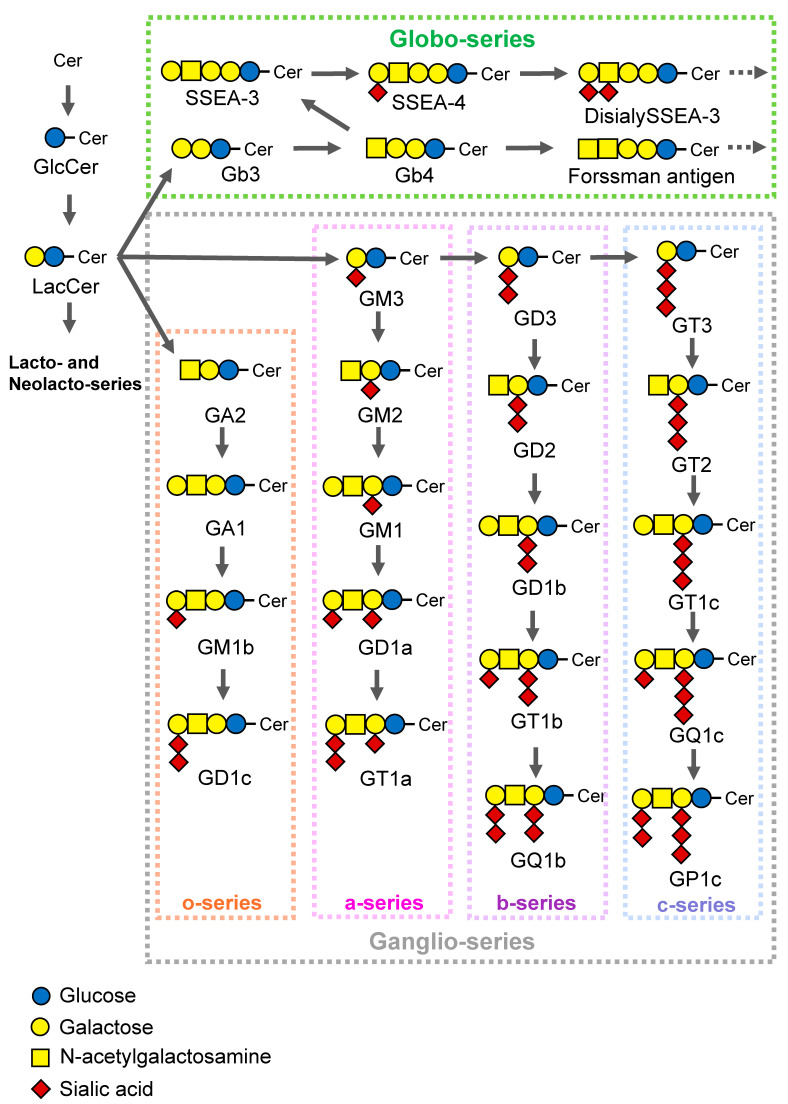
Synthetic pathway of glycosphingolipids.

**Figure 3 ijms-25-03568-f003:**
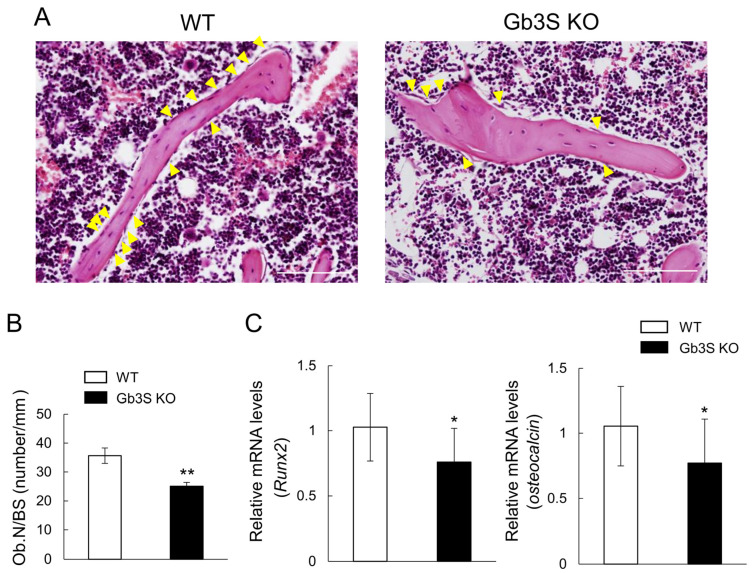
Decrease in osteoblast numbers and suppression of the gene expression of osteoblast differentiation markers by the deletion of Gb3 synthase (Gb3S KO). (**A**) Images of hematoxylin and eosin (HE) staining of femoral cancellous bone. Results for wild-type (WT) mice (**left**) and Gb3S KO mice (**right**) are shown. The scale bars are 100 μm. Yellow arrow heads indicate osteoblasts. (**B**) Osteoblast numbers/bone surface (Ob.N/BS, number/mm). (**C**) RNA isolated from the femur and tibia in WT and Gb3S KO mice. mRNA levels of *Runx2* (**left**) and *osteocalcin* (**right**). The single asterisk and double asterisks indicate *p* < 0.05 and *p* < 0.01, respectively. (**A**–**C**) are reproduced from our previous study [17].

**Figure 4 ijms-25-03568-f004:**
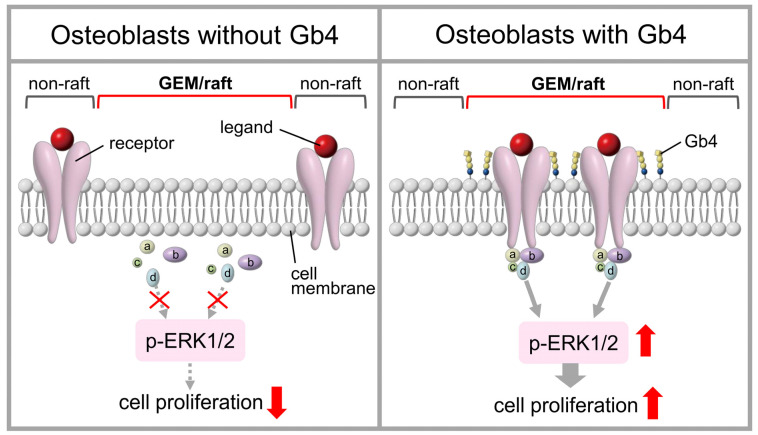
Schematic illustration of the proposed regulation of osteoblast proliferation by Gb4. Since receptors for proliferation exist in non-rafts in osteoblasts that do not express Gb4, the complexes (receptor, molecules of a, b, c, d) that transmit proliferation signals are not formed, which suppresses proliferation by inhibiting the phosphorylation of ERK1/2 (p-ERK1/2) (**left panel**). On the other hand, since receptors for proliferation are present in GEM/rafts in osteoblasts expressing Gb4, the complexes (receptor, molecules of a, b, c, d) that transmit proliferation signals are formed, resulting in the promotion of proliferation through increases in p-ERK1/2 (**right panel**).

**Figure 5 ijms-25-03568-f005:**
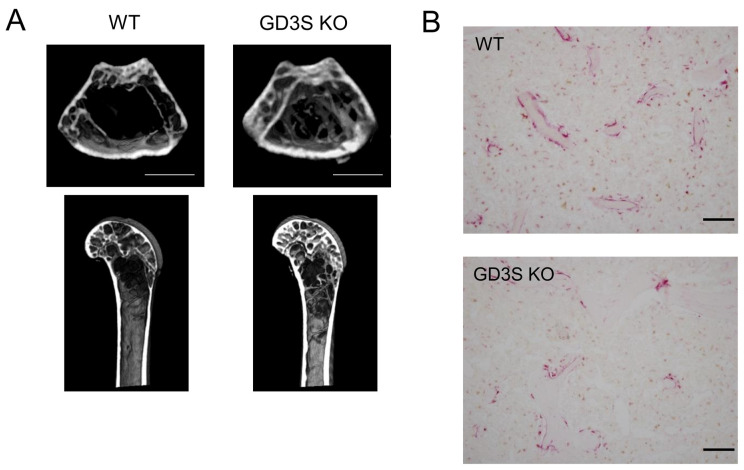
Decrease in the loss of femoral cancellous bone mass with aging in GD3 synthase knockout (GD3S KO) mice and reduction in the number of osteoclasts in femoral cancellous bone from GD3S KO mice. (**A**) µCT images of the femur in wild-type (WT) and GD3S KO mice at 40 weeks old. Results for WT mice (**left**) and GD3S KO mice (**right**) are shown. The scale bars are 1000 μm. (**B**) Images of the tartrate-resistant acid phosphatase (TRAP) staining of femoral cancellous bone. Results for WT mice (**upper**) and GD3S KO mice (**lower**) are shown. The scale bars are 100 μm. (**A**,**B**) are reproduced from our previous study [15].

**Figure 6 ijms-25-03568-f006:**
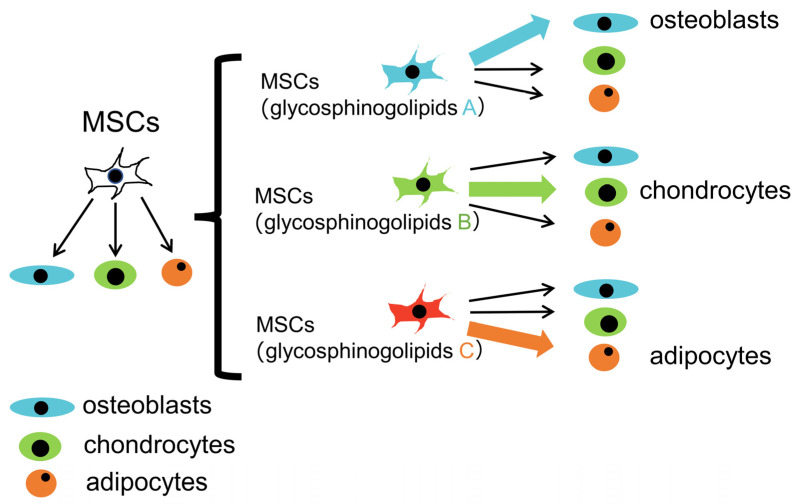
The differentiation tropism of MSCs due to differences in glycosphingolipids expressed in MSCs. MSCs with glycosphingolipids A are more likely to differentiate into osteoblasts, those with glycosphingolipids B are more likely to differentiate into chondrocytes, and those with glycosphingolipids C are more likely to differentiate into adipocytes.

## Data Availability

No new data were created or analyzed in this review. Data sharing is not applicable to this review.

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
