# Peer review of "Regulation of Glycosylation in Bone Metabolism"

_ijms, 2024, doi:10.3390/ijms25073568_

Round 1

Reviewer 1 Report

Comments and Suggestions for Authors

The authors herein reviewed the importance of the regulation of bone metabolism by glycoconjugates, such as glycolipids and glycoproteins, including the recent results.
This is a non-systematic review with an initial introduction where the role of glycosylation in different tissues and systems is considered with reference to bone.
Subsequently, the different types of glycosylation are classified, focusing in the next part on bone involvement. Its effect on osteoblasts and osteoclasts is analyzed.
Finally, its future role in bone research is considered.
The article provides information on the knowledge of glycosylation and bone

More specific comments:

1. The authors herein reviewed the importance of the regulation of bone metabolism by glycoconjugates, such as glycolipids and glycoproteins, including the recent results

2. This is a non-systematic review with an initial introduction where the role of glycosylation in different tissues and systems is considered with reference to bone.
Subsequently, the different types of glycosylation are classified, focusing in the next part on bone involvement. Its effect on osteoblasts and osteoclasts is analyzed.

3. The regulation osteoblast and osteoclast is very important to understand the physiopathology of bone metabolic disease. The role of glycosilation is not usually comment.

4. This is a non-systematic. Probably a systematic review with a described methodology could be interesting. This fact is a weakness

5.The conclusions are consistent with the review

6.The references are appropriate

7.The figures are very descriptive and will help you understand the mechanisms.

Reviewer 2 Report

Comments and Suggestions for Authors

Endogenous glycosylation involves a very broad scope of dynamic and complex processes that regulate numerous cellular activities.  This review focuses on the roles of glycolipids (glycosphingolipids) and glycoproteins in the development of osteoblasts and osteoclasts. Research progress in current literature was well summarized.  Two future directions regarding glycosphingolipids were proposed. Overall organization of the review is clear and easy to follow. This article may serve as a good guideline for researchers working in the fields of molecular glycoscience related to bone metabolism. Therefore, it is a good match for the journal.

 Minor points:

 1) It would be better to split the paragraph from line 94 to 134 on page 4 into two or more shorter paragraphs.  It is a little too long, therefore, not well focused.

2) There are a lot of abbreviations in this review.  It is hard not to miss one or two (such as RANKL in line 183, MYC in line 277, DNAX in line 300, just to list a few) that are not defined in the text.  I would suggest to have a separate standalone section for all the abbreviations mentioned in the review.

3) The sentence in line 287 may be changed to “Osteopontin on osteoclasts was also found to be N-glycosylated at the 79-position asparagine [62]”.

More specific comments:

This review focuses on the roles of glycolipids (glycosphingolipids) and glycoproteins in the development of osteoblasts and osteoclasts. It is relevant and interesting to researchers who are working in the fields of molecular glycoscience related to bone metabolism. Many reviews have been published on different specific topics of glycosylation because endogenous glycosylation involves a very broad scope of dynamic and complex processes that regulate numerous cellular activities. This review focuses on a niche area of bone metabolism, which is complementary to other reviews. This review is well written. Research progress in current literature was well summarized. Two future directions regarding glycosphingolipids were proposed. Overall organization of the review is clear and easy to follow. A concise section of conclusions was offered at the end of the review. It is short, but reflects the infant stage of the field.
